# FROM SCAN TO REAL DATA: SYSTEMATIC GENERALIZATION VIA MEANINGFUL LEARNING

## ABSTRACT

Humans can systematically generalize to novel compositions of existing concepts. There have been extensive conjectures into the extent to which neural networks can do the same. Recent arguments supported by evidence on the SCAN dataset claim that neural networks are inherently ineffective in such cognitive capacity. In this paper, we revisit systematic generalization from the perspective of meaningful learning, an exceptional capability of humans to learn new concepts by connecting them with other previously known knowledge. We propose to reassess models' compositional skills conditioned on the semantic connections between new and old concepts. In experiments, following the meaningful learning principle, we augment a training dataset in either an inductive or deductive manner to exposure such semantic links to models. Our observations on SCAN, as well as two real-world datasets on semantic parsing, suggest that modern sequence-to-sequence models, including RNNs, CNNs, and Transformers, can successfully one-shot generalize to novel concepts and compositions through semantic linking. We further demonstrate that both prior knowledge and semantic linking play a key role in achieving systematic generalization and that inductive learning generally works better than deductive learning. Lastly, we provide an explanation for data augmentation techniques by concluding them into either inductive-based or deductive-based meaningful learning. We hope our findings will encourage excavating existing neural networks' potential in systematic generalization through more advanced learning schemes.

## 1 INTRODUCTION

As a crucial characteristic of human cognition, systematic generalization reflects people's ability to learn infinite combinations of finite concepts (Chomsky, 1957; Montague et al., 1970). However, weak systematic compositionality has been considered as a primary obstacle to the expression of language and thought in connectionist networks for a long time (Fodor & Pylyshyn, 1988; Hadley, 1994; Marcus, 1998; Fodor & Lepore, 2002; Frank et al., 2009; Brakel & Frank, 2009; Marcus, 2018). Whether models can generalize systematically is still an appealing research topic until now. Recent works state that modern neural networks have not mastered these language-based generalization challenges in multiple explicitly proposed datasets (Lake & Baroni, 2017; Bastings et al., 2018; Keysers et al., 2019; Hupkes et al., 2020; Kim & Linzen, 2020). These studies conclude that models lack such cognitive capacity, which calls for a more systematic study. Apart from the proposal of benchmarks, existing research mainly focuses on novel architectural designs (Chen et al., 2020) data augmentation (Andreas, 2020; Akyürek et al., 2021) and meta-learning (Lake, 2019; Conklin et al., 2021) to enable systematic generalization.

In this work, however, we question that whether neural networks are indeed deficient or just conventional learning protocols unable to exploit their full potential (Csordás et al., 2021). Inspired by *meaningful learning* from the field of educational psychology (Mayer, 2002), we revisit systematic generalization to see whether neural networks still fail after *semantic linking*. To exposure semantic links between new concepts and existing ones, we augment prior knowledge through either *inductive learning* or *deductive learning* as what humans do in meaningful verbal learning (Ausubel, 1963). To be specific, inductive learning is a bottom-up approach from the more specific to the mode general. By introducing new concepts sharing the same context with existing ones in specific samples, we hope the model can capture the underlying semantic connections and thus generalize to novel compositions of new concepts. On the contrary, deductive learning is a top-down approach from the more general to the more specific. By involving a rule-like concept dictionary without specific

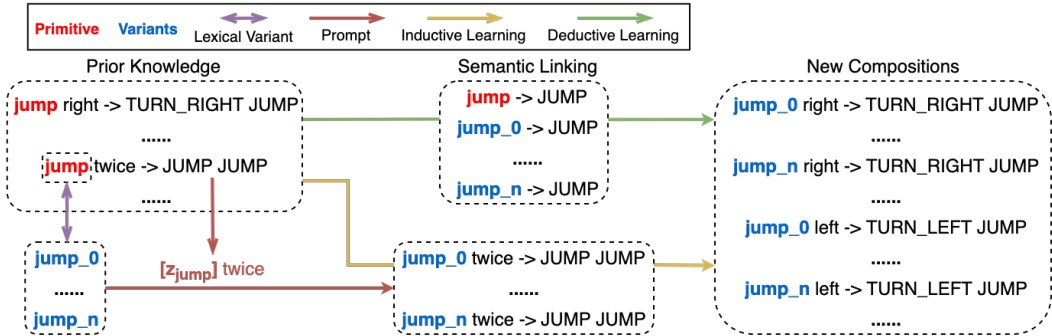

Figure 1: An illustration of the semantic linking injection pipeline in SCAN. The two middleboxes show the augmented dataset used for semantic linking through deductive learning (upper) and inductive learning (lower). In practice, the prior knowledge (left) and the augmented dataset (middle) are for training, and the new compositions of variants (right) are for testing. Models are expected to generalize to new compositions given prior knowledge and semantic linking.

context information, we hope the model can utilize the general cross-lingual supervised signals as anchor points so as to launch the semantic linking. In experiments, inductive and deductive learning stand for training models on extra samples with or without context, respectively, as two standard data augmentation techniques (Arthur et al., 2016; Wei & Zou, 2019; Nag et al., 2020). We mainly focus on three semantic relationships, namely, *lexical variant*, *co-hyponym*, and *synonym*. Starting from SCAN, our experiments confirm that, with semantic linking, even canonical neural networks can generalize systematically to new concepts and compositions. Moreover, this holds consistent across two more semantic parsing datasets. As an ablation study, we further examine such one-shot compositional generalization and find that both prior knowledge and semantic linking take essential parts. Lastly, we extend from toy sets to real data and explain how models' meaningful learning capability benefits them in solving real problems such as machine translation and semantic parsing with the assistance of data augmentation techniques.

Overall, our contributions are as follows: (1) We formally revisit systematic generalization by introducing semantic linking from a meaningful learning perspective. (2) We show how to conduct semantic linking by two common data augmentation approaches, either inductively or deductively. (3) We observe that modern neural networks can achieve systematic generalization with semantic linking, and both prior knowledge and semantic linking play a key role, which is in line with meaningful learning theory. (4) We extend from SCAN to real data and demonstrate that many recent data augmentation techniques belong to either inductive or deductive learning.

## 2 MEANINGFUL LEARNING

Learning new concepts by relating them to the existing ones is defined as a process of meaningful learning in educational psychology (Ausubel, 1963; Mayer, 2002). The utilization of meaningful learning can encourage learners to understand information continuously built on concepts the learners already understand (Okebukola & Jegede, 1988). Following the same idea, we intend to examine models' systematic compositionality by exploring semantic linking that establishes semantic relations between primitives $\mathbb{P}$ (old concepts) and their variants $\mathbb{V} := \{\mathbb{V}_p \mid \forall p \in \mathbb{P}\}$ (new concepts). To spoon-feed semantic knowledge to models for semantic linking, we augment the training data by either inductive learning or deductive learning (Hammerly, 1975; Shaffer, 1989; Thornbury, 1999) as humans learning vocabulary. In this section, we discuss the process of semantic linking and take "jump" from SCAN as an example primitive to illustrate the learning scheme in Figure 1.

### 2.1 SEMANTIC LINKS

We aim to revisit systematic generalization by exposing semantic links such as lexical variants, co-hyponyms, and synonyms. Lexical Variant refers to an alternative expression form for the same concept. Co-hyponym is a linguistic term to designate a semantic relation between two group members belonging to the same broader class, where each member is a hyponym and the class is a hypernym (Lyons & John, 1995). Synonym stands for a word, morpheme, or phrase that shares exactly or nearly the same semantics with another one. We provide an example and a detailed description in Appendix.

## 2.2 INDUCTIVE LEARNING

Inductive learning is a bottom-up approach from the more specific to the more general. In grammar teaching, inductive learning is a rule-discovery approach that starts with the presentation of specific examples from which a general rule can be inferred (Thornbury, 1999). Inspired by that, we propose to conduct semantic linking by introducing variant samples sharing the same context with their primitives during training. The assumption is that models can observe the interchange of primitives and their variants surrounded by the same context in the hope of coming up with a general hypothesis that there is a semantic linking between primitives and their variants (Harris, 1954). Formally, we describe inductive learning as follows. For a sequence-to-sequence task $\mathcal{T} : \boldsymbol{X} \rightarrow \boldsymbol{Y}$, we have a source sequence $\boldsymbol{x} \in \boldsymbol{X}$ and its target sequence $\boldsymbol{y} \in \boldsymbol{Y}$. We prepare prompts set $\boldsymbol{Z} := \{\boldsymbol{z} = f_{prompt}(\boldsymbol{x}) \mid \boldsymbol{x} \in \boldsymbol{X}\}$, where $f_{prompt}(\cdot)$ replaces the primitive in $\boldsymbol{x}$ with a slot mark $[z_p]$.[1] Then, we generate $\boldsymbol{X}^{IL} := \{\boldsymbol{x}^{IL} = f_{fill}(\boldsymbol{z}, v) \mid \boldsymbol{z} \in \boldsymbol{Z}, v \in \mathbb{V}\}$ by filling $[z_p]$ with variants in $\mathbb{V}_p$. There is no change from the target side, so we get $\boldsymbol{Y}^{IL}$ by copying $\boldsymbol{y}$ as $\boldsymbol{y}^{IL}$ for each $\boldsymbol{x}^{IL}$ correspondingly. Finally, we train models on $([\begin{smallmatrix} \boldsymbol{X} \\ \boldsymbol{X}^{IL} \end{smallmatrix}], [\begin{smallmatrix} \boldsymbol{Y} \\ \boldsymbol{Y}^{IL} \end{smallmatrix}])$ to operate semantic linking inductively. In practice, as shown in Figure 1, we first get a prompt "*[Z_{jump}] twice*" given a primitive sample "*jump twice*". After that, we generate variant samples by replacing the slot mask "*[Z_{jump}]*" with variants such as "*jump_0*". Finally, training models on generated variant samples like "*jump_0 twice*" combined with prior knowledge, we aim to establish the semantic relationships between primitives and their variants inductively. This process can be also treated as a kind of replacement augmentation (Wei & Zou, 2019).

## 2.3 DEDUCTIVE LEARNING

Deductive Learning, on the opposite of inductive learning, is a top-down approach from the more general to the more specific. As a rule-driven approach, teaching in a deductive manner often begins with presenting a general rule and is followed by specific examples in practice where the rule is applied (Thornbury, 1999). To align with this definition, we intend to do semantic linking deductively by combining a bilingual dictionary that maps primitives and their variants to the same in the target domain. This additional dictionary, hence, mixes the original training task with word translation (Mikolov et al., 2013b). Without any specific context, we hope the model can utilize the general cross-lingual supervised signals as anchor points so as to launch the semantic linking. Formally, we describe deductive learning as follows. We first treat $\mathbb{P}$ as the source dataset $\boldsymbol{X}_{\mathbb{P}}^{DL}$ directly and then prepare the corresponding target dataset $\boldsymbol{Y}_{\mathbb{P}}^{DL}$ by either decomposing samples from $\boldsymbol{Y}$ manually or feeding $\boldsymbol{X}_{\mathbb{P}}^{DL}$ to a trained external model. Similarly, we can consider $\mathbb{V}$ as another source dataset $\boldsymbol{X}_{\mathbb{V}}^{DL}$ and prepare its target dataset $\boldsymbol{Y}_{\mathbb{V}}^{DL}$ by copying the corresponding $\boldsymbol{y}_{\mathbb{P}}^{DL}$ as $\boldsymbol{y}_{\mathbb{V}}^{DL}$ for all $\boldsymbol{x}_{\mathbb{V}}^{DL}$ as variants of each $\boldsymbol{x}_{\mathbb{P}}^{DL}$. After all, we get $\boldsymbol{X}^{DL}$ as $[\begin{smallmatrix} \boldsymbol{X}_{\mathbb{P}}^{DL} \\ \boldsymbol{X}_{\mathbb{V}}^{DL} \end{smallmatrix}]$ and $\boldsymbol{Y}^{DL}$ as $[\begin{smallmatrix} \boldsymbol{Y}_{\mathbb{P}}^{DL} \\ \boldsymbol{Y}_{\mathbb{V}}^{DL} \end{smallmatrix}]$. The mapping from $\boldsymbol{X}^{DL}$ to $\boldsymbol{Y}^{DL}$ is a dictionary to translate primitives and their variants to the same targets without any specific context information. We name $(\boldsymbol{x}^{DL}, \boldsymbol{y}^{DL})$ as a *concept rule*, $(\boldsymbol{x}_{\mathbb{P}}^{DL}, \boldsymbol{y}_{\mathbb{P}}^{DL})$ as a *primitive rule*, and $(\boldsymbol{x}_{\mathbb{V}}^{DL}, \boldsymbol{y}_{\mathbb{V}}^{DL})$ as a *variant rule* since they are more rule-like without contexts. We train models on $([\begin{smallmatrix} \boldsymbol{X} \\ \boldsymbol{X}^{DL} \end{smallmatrix}], [\begin{smallmatrix} \boldsymbol{Y} \\ \boldsymbol{Y}^{DL} \end{smallmatrix}])$ to operate semantic linking deductively. In practice, as presented in Figure 1, we directly make use of primitive "*jump*" and its variants such as "*jump_0*" as the source sequences with action "JUMP" as their same target sequences. By exposing both the primitive rule "*jump*" → "JUMP" and variants rules like "*jump_0*"→ "JUMP" during training, we intend to build the semantic connections between primitives and their variants deductively. This manner is similar to the bilingual dictionary augmentation (Arthur et al., 2016; Nag et al., 2020).

## 3 SYSTEMATIC GENERALIZATION

Although previous studies argue that neural networks fail to match humans in systematic generalization (Lake & Baroni, 2017; Keysers et al., 2019), we revisit such algebraic compositionality conditioned on the semantic linking to see whether the conclusion will change. The following section moves on to specify the process and outcome of experiments. We first intend to make use of SCAN as the initial testbed to observe the presence of systematic generalization with the assistance of semantic relations. Then, we verify neural networks' potential to achieve the systematic generalization activated by semantic linking on SCAN, as well as two real-world tasks of semantic parsing. Following ablation studies further examine models' compositional capability.

---

[1]For each primitive, we pick only one prompt to fill in all its variants with $\boldsymbol{Z}$ specified for various datasets in Appendix.

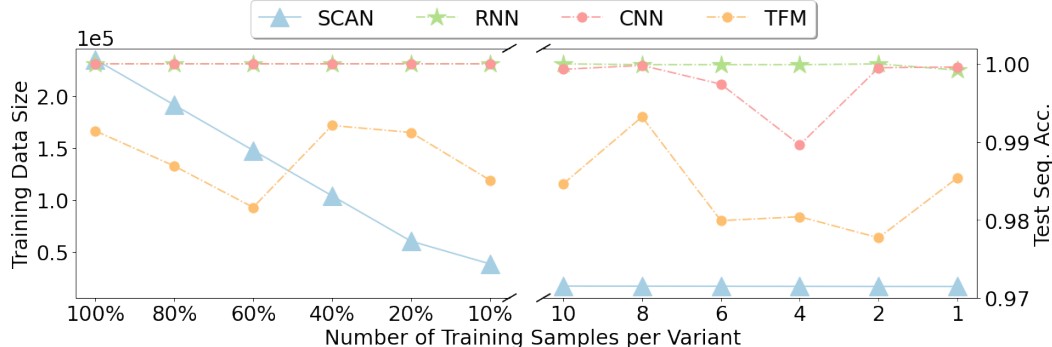

Figure 2: Experiments on SCAN with a decreasing number of training samples per variant from the complete set (100%) to a single sample (1). The solid line represents the change of overall training size, and the dashed line stands for that of test sequence accuracy. There is hardly a performance dip when training samples are deleted until only one remained.

## 3.1 DATASETS

There is evidence suggesting that SCAN may be far from enough to fully capture the kind of generalization, where even a simple model can behave as if it owns comparable skills (Bastings et al., 2018; Keysers et al., 2019). Thus, starting from SCAN, we introduce GEO and ADV generated respectively from real semantic parsing datasets: Geography and Advising.[2] Modification on datasets is specified in each experiment for the goal of examining machines' systematic generalization across various conditions.

**SCAN** is one of the benchmarks to investigate neural networks' compositional generalization (Lake & Baroni, 2017). It includes 20910 pairs of commands in English to their instructed action sequences [3]. We define $\mathbb{P}^{SCAN} := \{$ "*jump*", "*look*", "*run*", "*walk*" $\}$ to be in line with previous works. We focus on lexical variants and create $\mathbb{V}^{SCAN}$ by adding a suffix that consists of an underline and a unique number for each primitive. We control $|\mathbb{V}^{SCAN}|$ by setting the upper limit of this number. An example variant of "*jump*" is "*jump_0*" and both mean the same action "JUMP".

**Geography** is a common semantic parsing dataset (Zelle & Mooney, 1996; Srinivasan et al., 2017). It is also named as *geo880* since it contains 880 examples of queries about US geography in natural language paired with corresponding query expressions. It is later formatted to SQL language with variables in the target sequences (Finegan-Dollak et al., 2018). **GEO** is the dataset generated based on Geography, where we regard 4 of 9 annotated variables as hypernyms and keep them as they are in SQL sequences. The other variables are restored by entities from the source sequence accordingly. As a result, the overall data size is 618 after processing and we can make use of the "is-a" hypernymy relations between entities and variables for semantic linking. To be specific, we define $\mathbb{P}^{GEO} := \{$ "*new york city*", "*mississippi rivier*", "*dc*", "*dover*" $\}$ with $\mathbb{V}^{GEO}$ consisting of entities as co-hyponyms sharing the same variable group with primitives.[4] An example variant of "*new york city*" is "*houston city*" and both are in the same variable group "CITY_NAME".

**Advising**, as our second semantic parsing dataset, includes 4570 questions about course information in natural language paired with queries in SQL (Finegan-Dollak et al., 2018). Similar to GEO, **ADV** is generated on the basis of Advising with 4 of 26 variables as hypernyms. Precisely, we define $\mathbb{P}^{ADV} := \{$ "*a history of american film*", "*aaron magid*", "*aaptis*", "*100*" $\}$ and $\mathbb{V}^{ADV}$ as cohyponyms of primitives sharing the same variables. For instance, "*advanced at ai techniques*" is a co-hyponym of "*a history of american film*" sharing the same variable "TOPIC".

## 3.2 MODELS AND EXPERIMENTAL SETUP

What follows is an account of network configurations and experimental settings. Without specific instruction, they are shared throughout experiments.

**Models.** After testing a range of their adapted versions, we employ three dominant model candidates with an encoder-decoder framework (Sutskever et al., 2014), that is, RNN, CNN, and TFM. In terms

---

[2]https://github.com/jkkummerfeld/text2sql-data
[3]https://github.com/brendenlake/SCAN
[4]We select 4 primitives for both GEO and ADV to be align with SCAN.

of RNN, we reproduce bi-directional recurrent networks (Schuster & Paliwal, 1997) with long short-term memory units (Hochreiter & Schmidhuber, 1997) and an attention mechanism (Bahdanau et al., 2015). We follow the convolutional seq2seq architecture presented by Gehring et al. (2017) with regard to CNN and the attention-based structure proposed by Vaswani et al. (2017) in the case of TFM. More details are provided in Appendix.

**Training.** We apply the mini-batch strategy to sample 128 sequence pairs for each training step. We use Adam optimizer (Kingma & Ba, 2015) with an $\ell_2$ gradient clipping of 5.0 (Pascanu et al., 2013) and a learning rate of $1e^{-4}$ to minimize a cross-entropy loss. We freeze the maximum training epoch at 320 for CNN and 640 for RNN and TFM. In contrast to early stopping (Prechelt, 1998), we prefer a fixed training regime sufficient enough for models to fully converge in practice with a focus on the systematic generalization observation instead of superior structure exploration. To prevent uncontrolled interference, we train all models from scratch instead of fine-tuning (Devlin et al., 2019).

**Evaluation.** Token accuracy and sequence accuracy serve as two primary metrics in the following experiments. The former is a soft metric that allows partial errors in a sequence, while the latter is tricky and strictly does not. The reported results, along with standard deviation, are the mean of five runs.

### 3.3 EXPERIMENT: MEANINGFUL LEARNING

This experiment probes the models' compositional generalization via meaningful learning in SCAN. We compare performances across various conditions starting from the conventional training pipeline as a baseline. Usually, new concepts appear as out-of-vocabulary (OOV). A typical training pipeline often involves replacement (Wei & Zou, 2019) to handle new concepts, especially for those sharing same or close semantic with existing concepts. Thanks to their incredible algebraic compositionality, humans can effectively capture the underlying semantic connections between new concepts and old ones and generalize the prior knowledge to novel combinations by meaningful learning, given only a few demonstrations. To investigate the extent to which models can do the same, we gradually reduce the number of training samples generated by replacement augmentation until there is only one for each variant. Although the final one-shot learning (Vinyals et al., 2016) is challenging, we hope to observe the presence of models' meaningful learning by measuring the performance loss due to a decreasing number of training samples per variant.

**Experimental setup.** Following replacement augmentation, we assign placeholders at the positions of primitives in source sequences and later put back their 10 variants but keep the identical target sequences. Consequently, we have a total of 329,190 samples when $|\mathbb{V}^{SCAN}|$ is 40 and randomly split them into a training set (80%) and a test set (20%). The training set is further processed to remove samples having multiple variants. Thus, we ensure that the number of variants' occurrences is 1 while training at the one-shot condition. Eventually, the training dataset contains 235,002 samples. Models directly trained on this full dataset serve as baselines. Then, to format a gradual transition from baselines to meaningful learning, we train the same models on various datasets conditioned on a decreasing number of augmented samples for each variant until the one-shot learning setting. Besides, we use the variant rule "*jump_0*" → "JUMP" as the only training sample for "*jump_0*" as a case of our deductive learning and consider the rest as our inductive learning.

**Results.** Surprisingly, as elaborated in Figure 2, RNN has no significant performance drop when the training size is reduced from 235,002 (100%) to 16736 (1). It still achieves 99.92% test sequence accuracy when there is only one training sample for each variant. The same happens for CNN and TFM. Despite a slight fluctuation, they keep the results almost consistent regardless of whether the number of training variant samples is full or 1. The single sample works as a whole augmented dataset and enables models to generalize to novel compositions of learned variants. We want to underline that the sample can be either just a variant rule or a variant sample derived from a primitive prompt. As we defined, developing semantic linking through the former is deductive learning, and that through the latter is inductive learning. By utilizing such semantic relations between primitives and their variants, models show they can perform one-shot generalization systematically via meaningful learning after semantic linking.

### 3.4 SEMANTIC LINKING INJECTION

Having observed the success after semantic linking, one question that needs to be asked, however, is how it works. Therefore, the following two experiments evaluate models' systematic generalization,

Table 1: Dataset statistics in Section3.4. Test size is often dozens of times the training size due to replacement augmentation. Additional details are offered in Appendix.

| Data | SCAN | | | | | GEO | | | | | ADV | | | | |
|---|---|---|---|---|---|---|---|---|---|---|---|---|---|---|---|
| | Exp. IL | | | Exp. DL | | Exp. IL | | | Exp. DL | | Exp. IL | | | Exp. DL | |
| | Sta. | Dif. | Cha. | Sta. | Dif. | Sta. | Dif. | Cha. | Sta. | Dif. | Sta. | Dif. | Cha. | Sta. | Dif. |
| Train Size | 20946 | 20942 | 20928 | 20950 | 20946 | 724 | 720 | 711 | 728 | 724 | 6038 | 6034 | 5969 | 6040 | 6036 |
| Test Size | 308240 | 308240 | 308240 | 308240 | 308240 | 21350 | 21350 | 21350 | 21350 | 21350 | 107614 | 107614 | 107614 | 107614 | 107614 |

particularly for prior knowledge and semantic linking. A sliding scale of difficulty is carefully designed by weakening these two factors according to the policy that the greater the difficulty, the more compositional skills are required. We further validate our findings on GEO and ADV. We use the same evaluation protocol across different datasets in this section.

Given dataset $(\boldsymbol{X}, \boldsymbol{Y})$ as prior knowledge regarding primitives, we generate the test set by replacement augmentation. Specifically, we replace the primitives in source sequences with their variants to generate novel compositions. So far, variants exist as OOV since they are not in the training set. Then, we incorporate additional either $(\boldsymbol{X}^{IL}, \boldsymbol{Y}^{IL})$ or $(\boldsymbol{X}^{DL}, \boldsymbol{Y}^{DL})$ to the base training set $(\boldsymbol{X}, \boldsymbol{Y})$ so as to introduce variants in training and establish semantic linking inductively or deductively. As in the previous experiment, we ensure that each variant only has a single sample and appears only once. After training on $(\left[\begin{smallmatrix} \boldsymbol{X} \\ \boldsymbol{X}^{IL} \end{smallmatrix}\right], \left[\begin{smallmatrix} \boldsymbol{Y} \\ \boldsymbol{Y}^{IL} \end{smallmatrix}\right])$ or $(\left[\begin{smallmatrix} \boldsymbol{X} \\ \boldsymbol{X}^{DL} \end{smallmatrix}\right], \left[\begin{smallmatrix} \boldsymbol{Y} \\ \boldsymbol{Y}^{DL} \end{smallmatrix}\right])$, models are evaluated on the same test set prepared before by replacement augmentation. For convenience, we keep the same settings as in the previous experiment, where $|\mathbb{V}^{SCAN}|$ is 40 and 10 variants for each primitive. We use the full variants set for GEO, for example, 39 variants for "*new york city*", while we randomly sample 5 variants for each primitive in ADV so that we cover all the variants with an appropriate test size.

### 3.4.1 EXPERIMENT: SEMANTIC LINKING INJECTION VIA INDUCTIVE LEARNING

**Experimental setup.** We increase the difficulty of compositional learning by excluding primitive samples from the training set. We want to stress that, with a higher level of difficulty, models have to generalize not only to new concepts but also to their new compositions.

- **Standard**: Models are trained on $(\left[\begin{smallmatrix} \boldsymbol{X} \\ \boldsymbol{X}^{IL} \end{smallmatrix}\right], \left[\begin{smallmatrix} \boldsymbol{Y} \\ \boldsymbol{Y}^{IL} \end{smallmatrix}\right])$ without any adjustments.

- **Difficult**: We remove primitive samples sharing the same context with their variant samples. For example, we remove "*jump twice*" due to "*jump_0 twice*", and thus models have to generalize to "*jump_0 twice*" without seeing "*jump twice*".

- **Challenging:** We also exclude primitive training samples of the same length as their variant samples. For instance, models have to reproduce the same generalization to "*jump_0 twice*" without seeing primitive samples of length 2, including "*jump twice*", "*jump right*", "*jump left*", "*jump thrice*", and many others.[5]

**Results.** In SCAN, what stands out in Table 2 is an excellent one-shot generalization for all three networks. The participation of $(\boldsymbol{X}^{IL}, \boldsymbol{Y}^{IL})$ induces a near-perfect generalization. Even the worst results obtained by TFM in Challenging are around $98.76\%$ and $96.38\%$ in terms of token accuracy and sequence accuracy separately. The outcomes confirm that networks can inductively learn the semantic relation from context after semantic linking. After that, models of different architectures can successfully achieve systematic generalization to novel compositions of variants during the test. What is noticeable is a slight drop in both metrics as the difficulty upgrades. The disappearance of the training samples in Difficult and Challenging settings can lead to a performance drop. This is well in line with the widely accepted belief in meaningful learning theory, as well as our expectation, that prior knowledge is one of the keys related to humans' remarkable generalization. Therefore, we conclude that both semantic linking and background knowledge exert powerful effects upon the potential of models to generalize systematically. The trends above on SCAN can also be found on GEO and ADV, while more apparent changes in metrics again verify our findings that prior knowledge is essential. Either excluding primitive samples containing the same context or those of the same sequence length as their variant samples can produce a steep fall in the generalization, which is not so sharp on SCAN. On GEO, CNN can lose an absolute sequence accuracy of almost $18.26\%$ from Standard to Difficult, and that for TFM drops $7.66\%$. This upholds our argument that generalization via meaningful learning is inseparable from sufficient prior knowledge. The overall

---

[5]We only remove samples that will not lead to unknown tokens.

Table 2: Evaluation results over RNN, CNN, and TFM on SCAN, GEO, and ADV in Section 3.4.1 conditioned on Standard, Difficult and Challenging settings.

| Data | Model | Token Acc.% | | | Seq. Acc.% | | |
|------|-------|----------|-----------|-------------|----------|-----------|-------------|
| | | Standard | Difficult | Challenging | Standard | Difficult | Challenging |
| SCAN | RNN | $99.99 \pm 0.03$ | $99.89 \pm 0.19$ | $99.96 \pm 0.02$ | $99.95 \pm 0.08$ | $99.85 \pm 0.08$ | $99.80 \pm 0.31$ |
| | CNN | $99.96 \pm 0.08$ | $99.76 \pm 0.54$ | $98.89 \pm 2.44$ | $99.85 \pm 0.34$ | $99.52 \pm 1.07$ | $97.57 \pm 5.24$ |
| | TFM | $98.91 \pm 0.78$ | $98.90 \pm 1.10$ | $98.76 \pm 0.85$ | $97.35 \pm 1.62$ | $96.86 \pm 2.64$ | $96.38 \pm 2.81$ |
| GEO | RNN | $75.71 \pm 8.42$ | $75.69 \pm 6.12$ | $73.46 \pm 3.05$ | $44.95 \pm 14.69$ | $43.27 \pm 13.47$ | $36.77 \pm 5.60$ |
| | CNN | $87.99 \pm 2.67$ | $79.51 \pm 6.03$ | $77.40 \pm 2.48$ | $69.46 \pm 5.78$ | $51.20 \pm 8.64$ | $48.58 \pm 3.40$ |
| | TFM | $75.37 \pm 7.84$ | $75.11 \pm 4.88$ | $68.41 \pm 4.76$ | $45.93 \pm 12.42$ | $44.59 \pm 9.76$ | $36.93 \pm 7.47$ |
| ADV | RNN | $58.61 \pm 6.18$ | $59.74 \pm 5.67$ | $58.11 \pm 5.82$ | $36.18 \pm 5.75$ | $35.69 \pm 6.05$ | $35.45 \pm 6.69$ |
| | CNN | $57.83 \pm 7.55$ | $54.05 \pm 5.74$ | $53.66 \pm 2.57$ | $45.08 \pm 9.32$ | $42.14 \pm 6.90$ | $41.37 \pm 4.04$ |
| | TFM | $53.43 \pm 2.80$ | $51.51 \pm 4.50$ | $49.17 \pm 2.58$ | $42.59 \pm 3.65$ | $41.28 \pm 4.35$ | $38.88 \pm 2.68$ |

decline in performance can be attributed to the switch from toy sets to actual datasets since both GEO and ADV own a much more complex encoding and decoding space than SCAN.

### 3.4.2 Experiment: Semantic Linking Injection via Deductive Learning

**Experimental setup.** We increase the difficulty of compositional learning by excluding primitive rules from the training set as follows:

- **Standard**: Models are trained on $\left( \left[ \begin{smallmatrix} X \\ X^{DL} \end{smallmatrix} \right], \left[ \begin{smallmatrix} Y \\ Y^{DL} \end{smallmatrix} \right] \right)$ without any adjustments.

- **Difficult**: We remove primitive rules from the training set, and train models on $\left( \left[ \begin{smallmatrix} X \\ X_{\mathbb{V}}^{DL} \end{smallmatrix} \right], \left[ \begin{smallmatrix} Y \\ Y_{\mathbb{V}}^{DL} \end{smallmatrix} \right] \right)$.

**Results.** In SCAN, incorporating deductive semantic linking, all three networks are able to attain satisfying compositional generalization as shown in Table 3. CNN achieves the highest 99.96% in Standard, while TFM takes the lowest 91.26% in Difficult with regard to sequence accuracy. However, even the lowest one is impressive as there is only one variant rule to introduce each variant during training. We can see a consistent decline in accuracy across three different models if we undermine the semantic linking by removing primitive rules in Difficult. The most significant sequence accuracy drop of 3.3% came from CNN when the difficulty upgrades. We further validate our findings on GEO and ADV, and find a similar trend. There is a persistent performance loss because of the absence of primitive rules from the training set across models. Concretely in GEO, the grade of CNN declines from 32.33% in Standard to 23.58% in Difficult in terms of sequence accuracy. The causal role of semantic linking is also demonstrated by varying the difficulty. Overall, the joining of concept rules assists in developing semantic links between primitives and variants during training, by which models can behave compositionally during the test. Moreover, the different outcomes between Standard and Difficult indicate that either concept rules or just variant rules can connect primitives with their variants semantically, though the former is better than the latter. Again, the overall performance fall is the result of the more complicated task. Another noteworthy finding is that neural networks can realize systematic generalization in either an inductive or a deductive way but perform better in the former setting. By comparing such preference in Table 2 and Table 3, we find that current black-box neural nets are more capable of exploring rules and patterns from specific samples with context information rather than understanding knowledge from general concept rules in our experiments. This sheds light on why current machine learning is still highly data-driven and can hardly break through the bottleneck to realize advanced logic reasoning as human beings. How to improve models' generalization in deductive learning is an interesting future direction that we will work on.

### 3.5 Ablation Studies and Analysis

To explore other factors that may impact deductive learning, we conduct ablation studies with a varying $|\mathbb{P}^{SCAN}|$ from $\{1,2,3,4\}$ and $|\mathbb{V}^{SCAN}|$ from $\{1,5,10,15,20\}$ over RNN on SCAN. The experimental setup is borrowed from Standard in Section 3.4.2.

**Impact of $|\mathbb{P}^{SCAN}|$.** What attractive in Figure 3 (a) is that when the number of primitives grows, the generalization performance improves simultaneously in terms of both accuracy boosting and variance reduction. It is counter-intuitive to see such improvement as we expect that primitive rules should work independently, and the number of primitives should not impact the systematic

Table 3: Evaluation results over RNN, CNN, and TFM on SCAN, GEO, and ADV in Section 3.4.2 conditioned on Standard and Difficult settings.

| Data | Model | Token Acc.% | | Seq. Acc.% | |
|------|-------|-------------|-------------|-------------|-------------|
| | | Standard | Difficult | Standard | Difficult |
| SCAN | RNN | $99.48 \pm 0.71$ | $98.70 \pm 0.92$ | $98.27 \pm 2.38$ | $95.39 \pm 2.72$ |
| | CNN | $99.99 \pm 0.01$ | $98.59 \pm 3.10$ | $99.96 \pm 0.03$ | $96.66 \pm 7.27$ |
| | TFM | $96.90 \pm 1.78$ | $96.68 \pm 2.21$ | $91.94 \pm 4.04$ | $91.26 \pm 5.80$ |
| GEO | RNN | $54.44 \pm 7.15$ | $39.71 \pm 18.38$ | $13.61 \pm 7.08$ | $7.76 \pm 5.34$ |
| | CNN | $41.86 \pm 3.38$ | $41.07 \pm 7.48$ | $4.85 \pm 4.66$ | $4.04 \pm 2.18$ |
| | TFM | $67.02 \pm 6.91$ | $65.97 \pm 5.17$ | $36.38 \pm 10.08$ | $31.57 \pm 7.42$ |
| ADV | RNN | $36.50 \pm 7.66$ | $36.42 \pm 7.39$ | $12.84 \pm 4.31$ | $12.66 \pm 5.19$ |
| | CNN | $43.51 \pm 11.31$ | $35.34 \pm 14.68$ | $32.33 \pm 12.93$ | $23.58 \pm 16.04$ |
| | TFM | $56.82 \pm 3.79$ | $53.33 \pm 3.85$ | $47.43 \pm 3.71$ | $43.24 \pm 5.14$ |

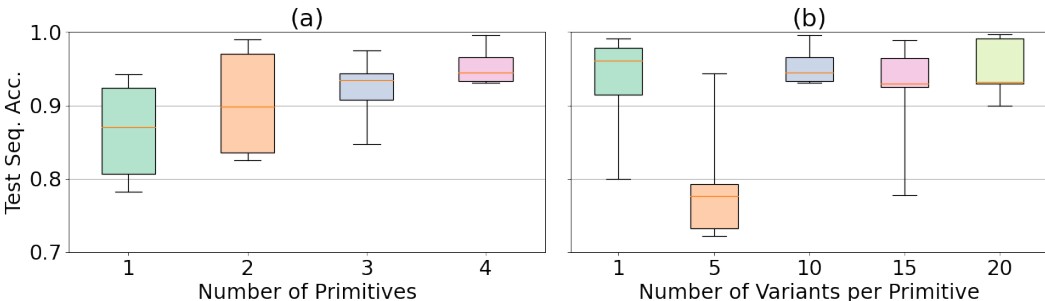

Figure 3: Experiments over RNN on SCAN with varying $|\mathbb{P}^{SCAN}|$ (a) and $|\mathbb{V}^{SCAN}|$ (b).

generalization a lot. A potential reason is that semantic linking built by various "independent" primitive rules can profit each other to trigger a more robust and stable systematic generalization. For example, "*jump*" $\rightarrow$ "JUMP" and "*look*" $\rightarrow$ "LOOK" separate them from the samples with context information, such as "***jump right***" and "***look right***". Thus, we can regard "*right*" as a compositional rules shared among primitive samples and finally encourage models to generalize effectively.

**Impact of** $|\mathbb{V}^{SCAN}|$**.** As presented in Figure 3 (b), RNN generalizes consistently well when the number of variants goes up. Therefore, we report that the generalization among variants of the same primitive has a certain degree of independence within a reasonable range (e.g., $\leq 20$).

## 4 FROM SCAN TO REAL DATA

Thus far, we have argued the feasibility of systematic generalization activated by semantic linking, as well as other factors such as prior knowledge. We move on now to discuss how such generalization already benefit the eventual performance of machines in solving real problems. Many recent papers propose to improve generalization on SCAN by data augmentation methods. Meta-learning is reported to solve compositional problems by equipping models with memory loading (Lake, 2019). The success is reasonable due to augmented data for the application of meta-learning. By considering concepts as pointers in the memory, models are designed to make connections between new and old concepts as semantic linking. Andreas (2020) suggests replacing fragments in real training samples with others that sharing similar contexts, which can also be supported by our inductive learning. As in our findings, similar context information can help establish the semantic links between new concepts and old ones, thus enable models to generalize to novel combinations. Besides, we have proved how replacement augmentation (Wei & Zou, 2019) works in Section 3.3. We would like to stress that the utility of similar unsupervised techniques (Xie et al., 2019) in both compositional generalization and real tasks can be attributed to inductive learning as well since there is a need for systematic generalization in practice.

In addition to inductive-based ones, we notice many works incorporating bilingual dictionaries (Arthur et al., 2016; Nag et al., 2020), or called concept rules by us, in low-resource machine translation can fall in the field of deductive-based methods. As a proof-of-concept, we reproduce the

Table 4: BLEU and SacreBLEU scores on IWSLT'14 English-German (En-De) and German-English (De-En), IWSLT'15 English-French (En-Fr) and French-English (Fr-En) translation tasks. We mark the addition of concept rules as *Vocabulary Augmentation*.

| Model | IWSLT'14 | | | | IWSLT'15 | | | |
| | En-De | | De-En | | En-Fr | | Fr-En | |
| | BLEU | SacreBLEU | BLEU | SacreBLEU | BLEU | SacreBLEU | BLEU | SacreBLEU |
|---|---|---|---|---|---|---|---|---|
| **Baselines** | | | | | | | | |
| LSTM (Luong et al., 2015) | 24.98 | 24.88 | 30.18 | 32.62 | 38.06 | 42.93 | 37.34 | 39.36 |
| Transformer (Vaswani et al., 2017) | 28.95 | 28.85 | 35.24 | 37.60 | 41.82 | 46.41 | 40.45 | 42.61 |
| Dynamic Conv. (Wu et al., 2019) | 27.39 | 27.28 | 33.33 | 35.54 | 40.41 | 45.32 | 39.61 | 41.42 |
| **+Vocabulary Augmentation** | | | | | | | | |
| LSTM (Luong et al., 2015) | $25.35\uparrow_{0.37}$ | $25.38\uparrow_{0.50}$ | $30.99\uparrow_{0.81}$ | $33.63\uparrow_{1.01}$ | $38.32\uparrow_{0.26}$ | $43.30\uparrow_{0.37}$ | $37.77\uparrow_{0.43}$ | $39.83\uparrow_{0.47}$ |
| Transformer (Vaswani et al., 2017) | $29.40\uparrow_{0.45}$ | $29.29\uparrow_{0.44}$ | $35.72\uparrow_{0.48}$ | $38.07\uparrow_{0.47}$ | $42.19\uparrow_{0.37}$ | $46.68\uparrow_{0.27}$ | $41.04\uparrow_{0.59}$ | $43.15\uparrow_{0.54}$ |
| Dynamic Conv. (Wu et al., 2019) | $27.60\uparrow_{0.21}$ | $27.50\uparrow_{0.22}$ | $33.62\uparrow_{0.29}$ | $36.00\uparrow_{0.46}$ | $40.87\uparrow_{0.46}$ | $45.95\uparrow_{0.63}$ | $39.95\uparrow_{0.34}$ | $41.86\uparrow_{0.44}$ |

Table 5: Token and sequence accuracy on Geography and Advising. We mark the addition of concept rules as *Entity Augmentation*.

| Model | Geography | | | | Advising | | | |
| | Train | | Test | | Train | | Test | |
| | Token Acc.% | Seq. Acc.% | Token Acc.% | Seq. Acc.% | Token Acc.% | Seq. Acc.% | Token Acc.% | Seq. Acc.% |
|---|---|---|---|---|---|---|---|---|
| **Baselines** | | | | | | | | |
| RNN | 89.05 | 17.39 | 69.81 | 9.68 | 92.22 | 3.64 | 60.41 | 6.11 |
| CNN | 98.45 | 70.74 | 78.44 | 55.91 | 99.74 | 81.62 | 81.74 | 51.13 |
| TFM | 99.45 | 84.95 | 80.24 | 49.82 | 99.68 | 76.90 | 78.51 | 29.67 |
| **+Entity Augmentation** | | | | | | | | |
| RNN | 87.47 | 29.96 | $72.39\uparrow_{2.58}$ | $15.05\uparrow_{5.37}$ | 88.82 | 30.97 | $71.17\uparrow_{10.76}$ | $16.06\uparrow_{9.95}$ |
| CNN | 97.54 | 76.03 | $80.32\uparrow_{1.88}$ | $60.93\uparrow_{5.02}$ | 99.65 | 87.01 | $84.50\uparrow_{2.76}$ | $56.02\uparrow_{4.89}$ |
| TFM | 99.30 | 85.73 | $81.09\uparrow_{0.85}$ | $54.84\uparrow_{5.02}$ | 99.57 | 86.94 | $84.26\uparrow_{5.75}$ | $35.08\uparrow_{5.41}$ |

word-to-word augmentation, or called deductive learning, by training models on not only the base training set but also concept rules. Intuitively, we wonder to which extent deductive semantic linking can promote models' performance in common machine translation (IWSLT'14 and IWSLT'150) and semantic parsing (Geography and Advising). In machine translation, we construct bilingual dictionaries by feeding vocabulary to the Google Translation API.[6] The word translation can be regarded as concept rules if we consider synonyms of a primitive as their variants and such synonymous relationships as semantic links. Consequently, we get 144,874 word samples as a training supplementary for En-De and De-En, and 110,099 for En-Fr and Fr-En. In semantic parsing, we construct entity dictionaries by collecting entities (e.g., "*new york city*"). They are translated to themselves since they do not change from the source natural language to the target SQL. The entity mapping can be treated as concept rules in the view of semantic linking. After that, we have a map of 103 entity translations for Geography and 1846 for Advising. A detailed experimental setup can be found in Appendix. We report the evaluation results in Table 4 and Table 5, where the same models with deductive semantic linking can consistently obtain performance gains.

## 5 CONCLUSION

Overall, we revisit systematic generalization from a meaningful learning perspective and introduce semantic linking to exposure semantic relations between new and old concepts to models during training. To establish such semantic networks, we take advantage of two common data augmentation methods and name them as inductive and deductive learning to align with the meaningful learning theory. The observed one-shot generalization on SCAN supports that neural networks as a class of modern machine learning methods can behave systematically after semantic linking. Extensive empirical results on SCAN, GEO, and ADV illustrate that prior knowledge and semantic linking are two essential factors in such generalization, in line with what humans do in meaningful learning. Given such findings, we further group recent data augmentation methods in either the inductive-based or deductive-based category, followed by a proof-of-concept to highlight how semantic linking already benefits models in solving real tasks such as machine translation and semantic parsing. We hope this paper can encourage future works to excavate neural networks' potential in systematic generalization through more advanced learning schemes.

---

[6] https://cloud.google.com/translate

ETHICS STATEMENT

As far as we know, we do not see any potential concerns such as negative societal impacts from our work.

REPRODUCIBILITY STATEMENT

Code, processed datasets, and experiment log of this work can be found in the attached supplementary materials and will be publicly available at GitHub. The source of raw datasets and external tools are presented in footnotes. Additional details regarding literature, model configurations, data statistics, and experimental setup are offered in Appendix as a reference in the main paper.

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

## A  APPENDIX

In the following pages, we discuss our work in detail. Our code and data can be found in the attached supplementary materials.

## B  SEMANTIC LINKS

**Lexical Variant**   refers to an alternative expression form for the same concept, where the various forms may derive from foreign languages, abbreviations, and even mistakes. A basic assumption is that all languages change over time due to non-linguistic factors. Since the rise of sociolinguistics in the 1960s, studies on linguistic variability, a characteristic of language, are central to the language use and motivations for speakers to vary the pronunciation, word choice, or morphology of existing concepts (Labov, 1963). Taking "*United States of America*" as an example, people have generally accepted the semantic connections among its lexical variants in history, including "*America*" and "*United States*", as well as the initialisms "*U.S.*" and "*U.S.A*". Many efforts have been devoted on lexical variants representation (Nguyen & Grieve, 2020), detection (Barteld, 2017), normalization (Baldwin et al., 2015) to keep machines up with the trend of the times.

**Co-hyponym**   is a linguistic term to designate a semantic relation between two group members belonging to the same broader class, where each member is a hyponym, also called subtype or subordinate, and the class is a hypernym (Lyons & John, 1995). The "is-a" hypernymy relation

between a generic hypernym and its specific hyponyms builds semantic connections among co-hyponyms. An example of such a hierarchical structure can be "*Mississippi*" and "*Massachusetts*" in the domain of "*state*". Specifically, "*Mississippi*" and "*Massachusetts*" are two hyponyms, and "*state*" is a hypernym. Thus, "*Mississippi*" and "*Massachusetts*" are semantically connected to be co-hyponyms for each other. Harvesting hypernymy relations (Wang & He, 2020) plays an essential role for downstream knowledge graph construction (Ji et al., 2021), out-vocabulary generalization (Dash et al., 2020), taxonomy expansion (Yu et al., 2020b), etc.

**Synonym** stands for a word, morpheme, or phrase that shares exactly or nearly the same semantics with another one. Many tend to assume synonyms are utterances that occur in most contexts in common, so they are semantically closely related enough to be synonyms for each other (Rubenstein & Goodenough, 1965; Harris, 1954). The existence of the association to contexts is a basic assumption supporting the advance of recent masked language modeling (Devlin et al., 2019). Given that, one of the definitions of a synonymous relation is a semantic link between two expressions if substitution of one for the other never hurts the true value of the context (Stanojević et al., 2009). For instance, the substitution of "*heavily populated*" for "*populous*" will seldom alter the truth of the sentence in Figure 4. Such semantic similarity can be observed in continuous vector space from a trained representation as well (Mikolov et al., 2013a). Synonym discovery (Yu et al., 2020a) has been a fundamental job to construct knowledge base and thus benefits substantial researches.

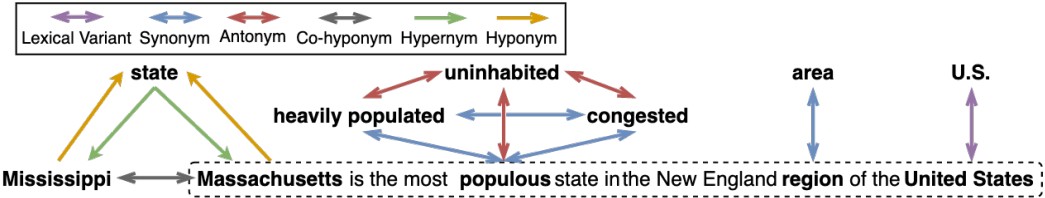

Figure 4: A concrete example of semantic linking. The bidirectional arrows denote symmetric relations. *Mississippi* and *Massachusetts* are two specific states, thus both hyponyms of *state*. In turn, *state* is a hypernym of them. Due to a common hypernym, *Mississippi* and *Massachusetts* become a co-hyponym for each other. {*heavily populated*, *congested*, *populus*}, {*area*, *region*} are two groups of synonyms for sharing same or similar semantics. Finally, *U.S.*, as a kind of abbreviation, is a lexical variant of *United States*.

## C  DATA

**IWSLT** involves IWSLT'14 (Cettolo et al., 2014) English-German (En-De) and German-English (De-En), IWSLT'15 (Cettolo et al., 2015) English-French (En-Fr) and French-English (Fr-En) translation tasks. The goal is to translate a sentence from one language to the other. The IWSLT'14 En-De and De-EN have 160,239 sequence pairs for training and 7,283 for validation. We make use of IWSLT14.TED.dev{2010, 2012} and IWSLT14.TED.tst{2010, 2011, 2012} to measure translation performance, resulting in a total of 6,750 test samples. In terms of IWSLT'15 En-Fr and Fr-En, there are 205,572 sequence pairs for training. We employ IWSLT15.TED.dev2010 and IWSLT15.TED.tst{2010, 2011, 2012, 2013} as the validation set and IWSLT15.tst{2014, 2015} as the test set. As a consequence, there are 5,519 samples for validation and 2,385 for evaluation. For all four translation tasks, we apply BPE with 10K tokens to share.

## D  MODELS

All models are developed with the encoder-decoder framework (Sutskever et al., 2014). We reproduce RNN, CNN, and TFM by ourselves to have fewer parameters than original versions for the convenience of verifying systematic generalization. The dropout rate is 0.5 for RNN, CNN, and TFM (Srivastava et al., 2014). We implement LSTM, Transformer, and Dynamic Conv. under the

Table 6: Example source and target sequences from SCAN, GEO, ADV, Geography, and Advising.

| Data | | Sequence |
|------|--------|----------|
| SCAN | Source | *jump twice* |
| | Target | JUMP JUMP |
| GEO | Source | *how many people in new york city* |
| | Target | SELECT CITY alias0 . POPULATION FROM CITY AS CITY alias0 WHERE CITY alias0 . CITY_NAME = CITY_NAME ; |
| ADV | Source | *Which department includes a history of american film ?* |
| | Target | SELECT DISTINCT COURSE alias0 . DEPARTMENT FROM COURSE AS COURSE alias0 WHERE COURSE alias0 . NAME LIKE TOPIC ; |
| Geography | Source | *how many people live in new york* |
| | Target | SELECT STATE alias0 . POPULATION FROM STATE AS STATE alias0 WHERE STATE alias0 . STATE_NAME = " new york " ; |
| Advising | Source | *I would like to see A History of American Film courses of 2 credits .* |
| | Target | SELECT DISTINCT COURSE alias0 . DEPARTMENT , COURSE alias0 . NAME , COURSE alias0 . NUMBER FROM COURSE AS COURSE alias0 WHERE ( COURSE alias0 . DESCRIPTION LIKE "% A History of American Film %" OR COURSE alias0 . NAME LIKE "% A History of American Film %" ) AND COURSE alias0 . CREDITS = 2 ; |

framework *fairseq*.[7] (Ott et al., 2019) and inherit its default model structures.[8] Without notes in tasks, hyperparameters are shared throughout the work. We train all of our models on a single Nvidia Tesla V100.

**RNN** denotes bi-directional recurrent network (Schuster & Paliwal, 1997; Hochreiter & Schmidhuber, 1997) with long short-term memory units and an attention mechanism (Bahdanau et al., 2015). Its encoder consists of two layers with a hidden size of 256 in each direction, and its decoder has one layer with a hidden size of 512. The embedding size is 512 for both encoder and decoder. There are a total of $5.29M$ trainable parameters. Teacher forcing with a rate of $0.5$ serves to spur up the training process (Williams & Zipser, 1989).

**CNN** denotes the fully convolutional seq2seq network (Gehring et al., 2017). The size of the position embedding layer is 128 for encoding and 256 for decoding, while that of the token embedding layer is 512 for both encoding and decoding. There are 10 convolutional layers with 512 as the hidden size and 3 as the kernel size in both encoder and decoder to generate a total of $33.55M$ trainable parameters.

**TFM** denotes transformers, an attention-based network (Vaswani et al., 2017). As a tiny version, TFM has 2 layers for each encoder and decoder with 8 attention heads and a dimension of 512. The size of the feedforward layer is 2048. We utilize the cyclic nature of $\sin$ and $\cos$ functions to represent token positions. There are a total of $15.02M$ trainable parameters.

**LSTM** is adapted from the recurrent network used by Luong et al. (2015) for statistical machine translation. The size of the embedding layer is 1000. There are 4 layers in both encoder and decoder with a hidden size of 512 and a dropout rate of 0.2.

**Transformer**, the same as TFM, is adapted from the base version of transformers in the work of Vaswani et al. (2017), while TFM is a tiny version to test systematic generalization. The dimension is 512 for the embedding layer, 1024 for the feedforward layer, and 512 for the attention layer. There are 6 attention blocks in both encoder and decoder with 4 attention heads and 0.3 dropout probability.

**Dynamic Conv.** is adapted from the seq2seq convolutional network proposed by Wu et al. (2019), where the hidden size of the embedding layer, encoder layer, and decoder layer is 512. The number of attention heads is 4, and the dimension of the feedforward layer is 1024 for both encoder and encoder. There are 6 layers in the encoder and 7 layers in the decoder. The dropout rate is 0.1 for both attention and weight units.

## E EXPERIMENTS

### SEMANTIC LINKING INJECTION VIA INDUCTIVE LEARNING

Semantic linking can be operated via inductive learning, where we replace the concept in the prompt with primitives and their variants. The learning rate to train CNN in GEO is changed to $5e^{-4}$.

---

[7]https://github.com/pytorch/fairseq

[8]LSTM is adapted from *lstm_luong_wmt_en_de*; Transformer is adapted from *transformer_iwslt_de_en*; Dynamic Conv. is adapted from *lightconv_iwslt_de_en*.

Table 7: Data statistics and training time per epoch in seconds. The batch size of each epoch for GEO and Geography is 32, and that for the others is 128.

| | SCAN | | | | | GEO | | | | | ADV | | | | | Geography | | Advising | |
|---|---|---|---|---|---|---|---|---|---|---|---|---|---|---|---|---|---|---|---|
| | Exp. 1 | | | Exp. 2 | | Exp. 1 | | | Exp. 2 | | Exp. 1 | | | Exp. 2 | | Bas. | Aug. | Bas. | Aug. |
| **Data** | Sta. | Dif. | Cha. | Sta. | Dif. | Sta. | Dif. | Cha. | Sta. | Dif. | Sta. | Dif. | Cha. | Sta. | Dif. | | | | |
| **Train Size** | 20946 | 20942 | 20928 | 20950 | 20946 | 724 | 720 | 711 | 728 | 724 | 6038 | 6034 | 5969 | 6040 | 6036 | 598 | 701 | 3814 | 5660 |
| **Test Size** | 308240 | 308240 | 308240 | 308240 | 308240 | 21350 | 21350 | 21350 | 21350 | 21350 | 107614 | 107614 | 107614 | 107614 | 107614 | 279 | 279 | 573 | 573 |
| **Time** RNN | | | 21 | | | | | 5 | | | | | 19 | | | 4 | 5 | 27 | 35 |
| CNN | | | 17 | | | | | 1.2 | | | | | 11 | | | 1 | 1.2 | 12 | 19 |
| TFM | | | 7 | | | | | 0.5 | | | | | 5 | | | 0.4 | 0.5 | 6 | 8 |

Table 8: Prompts with example primitives and sampled variants. In SCAN, primitives share the same prompt and the number of variants can be changed. In ADV, we randomly sample 5 variants for each source sequence so that we cover all the variants with a test set of an appropriate size.

| Data | Primitive | Variant | #Variants | Template |
|---|---|---|---|---|
| SCAN | *jump* | *jump_0* | 10 | *[concept] twice* |
| GEO | *new york city* | *houston city* | 39 | *how many people in [concept]* |
| | *mississippi rivier* | *red rivier* | 9 | *how long is [concept]* |
| | *dc* | *kansas* | 49 | *where is [concept]* |
| | *dover* | *salem* | 8 | *what states capital is [concept]* |
| ADV | *a history of american film* | *advanced ai techniques* | 5/424 | *who teaches [concept] ?* |
| | *aaron magid* | *cargo* | 5/492 | *does [concept] give upper-level courses ?* |
| | *aaptis* | *survmeth* | 5/1720 | *name core courses for [concept] .* |
| | *100* | *171* | 5/1895 | *can undergrads take [concept] ?* |

Prompts used in SCAN, GEO, and ADV are expressed in Table 8. Detailed experimental results with respect to three levels can be found in Table 9, Table 10, and Table 11.

Table 9: Results of Standard inductive learning.

| Data | Model | Train | | | Test | | |
|---|---|---|---|---|---|---|---|
| | | **Loss** | **Token Acc.%** | **Seq. Acc.%** | **Loss** | **Token Acc.%** | **Seq. Acc.%** |
| SCAN | RNN | $0.00 \pm 0.00$ | $100.00 \pm 0.00$ | $99.99 \pm 0.02$ | $0.00 \pm 0.00$ | $99.99 \pm 0.03$ | $99.95 \pm 0.08$ |
| | CNN | $0.00 \pm 0.00$ | $99.81 \pm 0.09$ | $98.78 \pm 0.55$ | $0.00 \pm 0.00$ | $99.96 \pm 0.08$ | $99.85 \pm 0.34$ |
| | TFM | $0.00 \pm 0.00$ | $99.82 \pm 0.02$ | $98.83 \pm 0.12$ | $0.06 \pm 0.03$ | $98.91 \pm 0.78$ | $97.35 \pm 1.62$ |
| GEO | RNN | $0.15 \pm 0.02$ | $97.73 \pm 0.42$ | $80.25 \pm 2.81$ | $1.36 \pm 0.48$ | $75.71 \pm 8.42$ | $44.95 \pm 14.69$ |
| | CNN | $0.07 \pm 0.01$ | $98.23 \pm 0.39$ | $76.80 \pm 2.25$ | $9.01 \pm 4.26$ | $87.99 \pm 2.67$ | $69.46 \pm 5.78$ |
| | TFM | $0.02 \pm 0.00$ | $99.63 \pm 0.07$ | $91.60 \pm 1.41$ | $4.55 \pm 1.39$ | $75.37 \pm 7.84$ | $45.93 \pm 12.42$ |
| ADV | RNN | $0.03 \pm 0.01$ | $99.40 \pm 0.13$ | $82.74 \pm 2.78$ | $6.04 \pm 0.95$ | $58.61 \pm 6.18$ | $36.18 \pm 5.75$ |
| | CNN | $0.01 \pm 0.01$ | $99.59 \pm 0.07$ | $85.13 \pm 1.95$ | $23.56 \pm 4.95$ | $57.83 \pm 7.55$ | $45.08 \pm 9.32$ |
| | TFM | $0.00 \pm 0.00$ | $99.92 \pm 0.01$ | $96.14 \pm 0.28$ | $15.12 \pm 1.00$ | $53.43 \pm 2.80$ | $42.59 \pm 3.65$ |

Table 10: Results of Difficult inductive learning.

| Data | Model | Train | | | Test | | |
|---|---|---|---|---|---|---|---|
| | | **Loss** | **Token Acc.%** | **Seq. Acc.%** | **Loss** | **Token Acc.%** | **Seq. Acc.%** |
| SCAN | RNN | $0.00 \pm 0.00$ | $100.00 \pm 0.00$ | $99.99 \pm 0.01$ | $0.00 \pm 0.00$ | $99.96 \pm 0.02$ | $99.85 \pm 0.08$ |
| | CNN | $0.00 \pm 0.00$ | $99.77 \pm 0.19$ | $98.62 \pm 1.13$ | $0.03 \pm 0.06$ | $99.76 \pm 0.54$ | $99.52 \pm 1.07$ |
| | TFM | $0.00 \pm 0.00$ | $99.79 \pm 0.03$ | $98.59 \pm 0.12$ | $0.06 \pm 0.03$ | $98.90 \pm 1.10$ | $96.86 \pm 2.64$ |
| GEO | RNN | $0.16 \pm 0.03$ | $97.39 \pm 0.67$ | $78.33 \pm 4.31$ | $1.29 \pm 0.27$ | $75.69 \pm 6.12$ | $43.27 \pm 13.47$ |
| | CNN | $0.07 \pm 0.01$ | $98.25 \pm 0.13$ | $76.53 \pm 1.68$ | $13.87 \pm 3.19$ | $79.51 \pm 6.03$ | $51.20 \pm 8.64$ |
| | TFM | $0.00 \pm 0.11$ | $99.60 \pm 0.11$ | $91.33 \pm 1.46$ | $4.50 \pm 0.80$ | $75.11 \pm 4.88$ | $44.59 \pm 9.76$ |
| ADV | RNN | $0.03 \pm 0.01$ | $99.26 \pm 0.21$ | $79.57 \pm 4.12$ | $5.80 \pm 0.92$ | $59.74 \pm 5.67$ | $35.69 \pm 6.05$ |
| | CNN | $0.02 \pm 0.00$ | $99.56 \pm 0.05$ | $84.06 \pm 1.57$ | $24.58 \pm 3.40$ | $54.05 \pm 5.74$ | $42.14 \pm 6.90$ |
| | TFM | $0.00 \pm 0.00$ | $99.91 \pm 0.01$ | $95.88 \pm 0.23$ | $15.84 \pm 1.51$ | $51.51 \pm 4.50$ | $41.28 \pm 4.35$ |

Table 11: Results of Challenging inductive learning.

| Data | Model | Train | | | Test | | |
|------|-------|-------|---|---|------|---|---|
| | | Loss | Token Acc.% | Seq. Acc.% | Loss | Token Acc.% | Seq. Acc.% |
| SCAN | RNN | $0.00 \pm 0.00$ | $100.00 \pm 0.00$ | $99.99 \pm 0.02$ | $0.20 \pm 0.45$ | $99.95 \pm 0.08$ | $99.80 \pm 0.31$ |
| | CNN | $0.00 \pm 0.00$ | $99.85 \pm 0.05$ | $99.00 \pm 0.30$ | $0.14 \pm 0.31$ | $98.89 \pm 2.44$ | $97.57 \pm 5.24$ |
| | TFM | $0.00 \pm 0.00$ | $99.82 \pm 0.05$ | $98.85 \pm 0.27$ | $0.07 \pm 0.05$ | $98.76 \pm 0.85$ | $96.38 \pm 2.81$ |
| GEO | RNN | $0.15 \pm 0.04$ | $97.76 \pm 0.74$ | $79.77 \pm 4.19$ | $1.52 \pm 0.29$ | $73.46 \pm 3.05$ | $36.77 \pm 5.60$ |
| | CNN | $0.07 \pm 0.01$ | $98.23 \pm 0.17$ | $75.98 \pm 1.46$ | $15.83 \pm 4.56$ | $77.40 \pm 2.48$ | $48.53 \pm 3.40$ |
| | TFM | $0.02 \pm 0.00$ | $99.60 \pm 0.06$ | $91.00 \pm 1.20$ | $6.01 \pm 1.03$ | $68.41 \pm 4.76$ | $36.93 \pm 7.47$ |
| ADV | RNN | $0.03 \pm 0.01$ | $99.23 \pm 0.13$ | $79.90 \pm 1.85$ | $5.95 \pm 0.90$ | $58.11 \pm 5.82$ | $35.45 \pm 6.69$ |
| | CNN | $0.01 \pm 0.01$ | $99.68 \pm 0.15$ | $87.90 \pm 5.05$ | $23.08 \pm 6.34$ | $53.66 \pm 2.57$ | $41.37 \pm 4.04$ |
| | TFM | $0.00 \pm 0.00$ | $99.93 \pm 0.01$ | $96.41 \pm 0.24$ | $16.59 \pm 0.98$ | $49.17 \pm 2.58$ | $38.88 \pm 2.68$ |

Table 12: Concept rules with primitives and their example variants.

| Data | Primitive | Semantic Links | Variant | Concept Rule | |
|------|-----------|----------------|---------|--------------|---|
| | | | | Primitive Rule | Variant Rule |
| SCAN | *jump* *look* *run* *walk* | Lexical Variant | *jump_0* *look_0* *run_0* *walk_0* | *jump* → JUMP *look* → LOOK *run* → RUN *walk* → WALK | *jump_0* → JUMP *look_0* → LOOK *run_0* → RUN *walk_0* → WALK |
| GEO | *new york city* *mississippi rivier* *dc* *dover* | Co-hyponym | *houston city* *red rivier* *kansas* *salem* | *new york city* → CITY_NAME *mississippi rivier* → RIVER_NAME *dc* → STATE_NAME *dover* → CAPITAL_NAME | *houston city* → CITY_NAME *red rivier* → RIVER_NAME *kansas* → STATE_NAME *salem* → CAPITAL_NAME |
| ADV | *a history of american film* *aaron magid* *aaptis* *100* | Co-hyponym | *advanced ai techniques* *cargo* *survmeth* *171* | *a history of american film* → TOPIC *aaron magid* → INSTRUCTOR *aaptis* → DEPARTMENT *100* → NUMBER | *advanced ai techniques* → TOPIC *cargo* → INSTRUCTOR *survmeth* → DEPARTMENT *171* → NUMBER |

SEMANTIC LINKING INJECTION VIA DEDUCTIVE LEARNING

Semantic linking can be established via deductive learning, where we put concept rules without context information in the training set instead of specific sequence samples. Example concept rules for SCAN, GEO, and ADV are presented in Table 12. Detailed experimental results with respect to two levels can be found in Table 13 and Table 14.

Table 13: Results of Standard deductive learning.

| Data | Model | Train | | | Test | | |
|------|-------|-------|---|---|------|---|---|
| | | Loss | Token Acc.% | Seq. Acc.% | Loss | Token Acc.% | Seq. Acc.% |
| SCAN | RNN | $0.00 \pm 0.00$ | $99.99 \pm 0.03$ | $99.90 \pm 0.23$ | $0.05 \pm 0.06$ | $99.48 \pm 0.71$ | $98.27 \pm 2.38$ |
| | CNN | $0.00 \pm 0.00$ | $99.79 \pm 0.14$ | $98.78 \pm 0.79$ | $0.00 \pm 0.00$ | $99.99 \pm 0.01$ | $99.96 \pm 0.03$ |
| | TFM | $0.00 \pm 0.00$ | $99.82 \pm 0.03$ | $98.78 \pm 0.17$ | $0.27 \pm 0.22$ | $96.90 \pm 1.78$ | $91.94 \pm 4.04$ |
| GEO | RNN | $0.17 \pm 0.03$ | $97.50 \pm 0.30$ | $78.54 \pm 2.16$ | $2.83 \pm 0.69$ | $54.44 \pm 7.15$ | $13.61 \pm 7.08$ |
| | CNN | $0.08 \pm 0.01$ | $97.97 \pm 0.24$ | $77.03 \pm 1.42$ | $51.08 \pm 25.97$ | $41.86 \pm 3.38$ | $4.85 \pm 4.66$ |
| | TFM | $0.02 \pm 0.00$ | $99.54 \pm 0.31$ | $91.82 \pm 2.27$ | $6.03 \pm 1.56$ | $67.02 \pm 6.91$ | $36.38 \pm 10.08$ |
| ADV | RNN | $0.08 \pm 0.02$ | $98.64 \pm 0.31$ | $68.84 \pm 4.57$ | $7.95 \pm 1.13$ | $36.50 \pm 7.66$ | $12.84 \pm 4.31$ |
| | CNN | $0.02 \pm 0.00$ | $99.53 \pm 0.07$ | $84.64 \pm 1.20$ | $31.12 \pm 4.76$ | $43.51 \pm 11.31$ | $32.33 \pm 12.93$ |
| | TFM | $0.00 \pm 0.00$ | $99.91 \pm 0.02$ | $96.33 \pm 0.37$ | $13.72 \pm 1.41$ | $56.82 \pm 3.79$ | $47.43 \pm 3.71$ |

MACHINE TRANSLATION

We show how semantic linking already benefits models' performance in machine translation. The semantic links between primitives and their variants in machine translation is built upon the synonymous relations between tokens such as "*heavily populated*" and "*populous*". Given that synonymous connection is reversible as shown in Figure 4, a primitive can also be the other primitives' variant. Specifically, we collect a dictionary of tokens for the source language and feed the token to the Google Translation API to obtain a token map from the source language to the target one. The same operation can be repeated from the target language to the source one. Two dictionaries are combined into one with duplicates removed. Consequently, we get 144,874 token-level samples as a

Table 14: Results of Difficult deductive learning.

| Data | Model | Train | | | Test | | |
|------|-------|------|-------------|------------|------|-------------|------------|
| | | Loss | Token Acc.% | Seq. Acc.% | Loss | Token Acc.% | Seq. Acc.% |
| SCAN | RNN | $0.00 \pm 0.00$ | $99.99 \pm 0.01$ | $99.95 \pm 0.07$ | $0.08 \pm 0.08$ | $98.70 \pm 0.92$ | $95.39 \pm 2.72$ |
| | CNN | $0.00 \pm 0.00$ | $99.62 \pm 0.34$ | $98.82 \pm 1.09$ | $0.13 \pm 0.29$ | $98.59 \pm 3.10$ | $96.66 \pm 7.27$ |
| | TFM | $0.00 \pm 0.00$ | $99.82 \pm 0.03$ | $98.78 \pm 0.12$ | $0.21 \pm 0.20$ | $96.68 \pm 2.21$ | $91.26 \pm 5.80$ |
| GEO | RNN | $0.20 \pm 0.03$ | $96.93 \pm 0.71$ | $75.35 \pm 3.57$ | $4.40 \pm 2.50$ | $39.71 \pm 18.38$ | $7.67 \pm 5.34$ |
| | CNN | $0.08 \pm 0.01$ | $97.77 \pm 0.76$ | $76.41 \pm 2.80$ | $32.94 \pm 4.26$ | $41.07 \pm 7.48$ | $4.04 \pm 2.18$ |
| | TFM | $0.02 \pm 0.00$ | $99.56 \pm 0.11$ | $91.08 \pm 1.56$ | $5.97 \pm 1.05$ | $65.97 \pm 5.17$ | $31.57 \pm 7.42$ |
| ADV | RNN | $0.08 \pm 0.02$ | $98.54 \pm 0.28$ | $67.10 \pm 3.45$ | $7.87 \pm 1.01$ | $36.42 \pm 7.39$ | $12.66 \pm 5.19$ |
| | CNN | $0.04 \pm 0.05$ | $98.78 \pm 1.91$ | $77.14 \pm 23.28$ | $32.44 \pm 6.07$ | $35.34 \pm 14.68$ | $23.58 \pm 16.04$ |
| | TFM | $0.00 \pm 0.00$ | $99.92 \pm 0.02$ | $96.41 \pm 0.26$ | $14.92 \pm 1.31$ | $53.33 \pm 3.85$ | $43.24 \pm 5.14$ |

training supplementary for IWSLT'14 En-De and De-En, and 110,099 for IWSLT'15 En-Fr and Fr-En, which leads to a total of 305,113 training samples for IWSLT'14 En-De and De-En and 315,671 for IWSLT'15 En-Fr and Fr-En after such vocabulary augmentation.

**Experimental Setup.** We evaluate our approach on IWSLT'14 Cettolo et al. (2014) English-German (En-De) and German-English (De-En), IWSLT'15 Cettolo et al. (2015) English-French (En-Fr) and French-English (Fr-En) translation tasks. We follow the standard evaluation protocol Ott et al. (2019) that keeps the original training set and validation set but combines multiple previous test sets for final evaluation[9]. We apply BPE with 10K tokens for all tasks and report both BLEU Papineni et al. (2002) and SacreBLEU Post (2018) scores for three baselines: LSTM Luong et al. (2015), Transformer Vaswani et al. (2017), and Dynamic Conv. Wu et al. (2019) in comparsion with same structures augmented by our method.

**Results.** From Table 4, we observe a consistent improvement in both BLEU and SacreBLEU over all baselines when performed vocabulary augmentation, particularly up to 1 in SacreBlEU. The additional synonym pairs not only construct the semantic linking between tokens in two languages explicitly, but also create a complicated semantic linking network implicitly because of synonyms within the single language and the transitivity nature of synonym relation. Our experiments prove that semantic linking, which allows models to generalize systematically, is also beneficial for improving machine translation performance.

SEMANTIC PARSING

We consider a variable as a hypernym for its values and entities belonging to the same variable as co-hyponyms in semantic parsing. Thus, we can treat all the entity values for all variables as primitives and the translations from primitives to their variables as primitive rules that later joins the base training set. For a fair comparison, a token from this extra dataset will be marked as a unique unknown mark, "¡unk¿", if it does not exist in the original base training set. After that, we have a map of 103 entity translations for Geography and 1846 for Advising, resulting in a training size change from 701 to 804 for Geography and from 3814 to 5660 for Advising.

**Experimental Setup.** We evaluate our method on two semantic parsing benchmarks, Geography, and Advising. We train the same models as we analyzed before without more hyperparameter tuning, including RNN, CNN, and TFM. There are some changes for CNN, where the learning rate is $5e^{-4}$ in Geography, and the maximum sequence length for the decoder position embedding is 312 in Advising. We split $10\%$ training samples as the validation set to find the converged epoch and then add it back to the training set for the final report.

**Results.** As elaborated in Table 5, all three networks can achieve better performance in terms of both accuracy and variance. Specifically, a $10.76\%$ token accuracy and $9.95\%$ sequence accuracy boosting are observed from RNN on Advising after such entity augmentation. The results suggest that models can learn semantic linking or be more familiar with similar contexts from those primitive rules in a deductive way to enhance model systematic generalization and finally lead to better outcomes.

---

[9]The final test set of IWSLT'14 consists of IWSLT14.TED.dev{2010, 2012} and IWSLT14.TED.tst{2010, 2011, 2012} ; and IWSLT'15 includes IWSLT15.TED.tst{2014, 2015} Ott et al. (2019).

