# OpenReview forum: "From SCAN to Real Data: Systematic Generalization via Meaningful Learning"
_ICLR.cc/2022/Conference — ICLR 2022 Submitted_

### Official Review · Reviewer_yQN2 · 2021-10-29

**Correctness:** 3
**Technical Novelty And Significance:** 1
**Empirical Novelty And Significance:** 2
**Recommendation:** 5
**Confidence:** 4

**Main Review:**

Merits:
* This paper focuses on an important problem - systematic genneralization and compositional generalization in particular.
* The experimental evaluation is sound including both synthetic and NMT tasks. The evaluation protocal and the design of three tasks are valuable for followups.
* The empirical performances are encouraging and interesting.

Flaws:
* The model depends heavily on the prior knowledge about the structure of the task but essentially a more realistic approach towards compositionality is concerning how to induce gramatical rules over raw natural language (considering the birttleness of many neural-symbolic systems).
* Considering there are already many works using data augmentation to improved compositional generalizaiton [1,2], the algorithmic novelty only combining with semantic linking is quite doubleful in my opinion. It is interesting to unify existing (data-augmentation) frameworks but the efforts are not solid enough.
* The inductive and deductive processes are not very well justified. For example, in symbolism, inductive learning induce (logical) rules explicitly by composing predicates. But in the paper, it seems the model only use rules to generate extra parrallel data for training.
* It would be nice to discuss more on the connections between the intruiging performance in synthetic dataset like SCAN and the systematic generalization capability over real-world cases [3, 4]. Considering there are many works on compositional
whether they are only exploiting dataset biases is an open question and suits the title of this work.
* The claims in the paper are a bit vague like 'meaningful learning' (seems like a cognitive concept but not very clear to many authors). And the experiment part is well-written but the previous parts are not as good as it. It would be much better to have an algorithm for introducing the method.

[1] Learning to Recombine and Resample Data for Compositional Generalization

[2] Good-Enough Compositional Data Augmentation

[3] The Devil is in the Detail: Simple Tricks Improve Systematic Generalization of Transformers

[4] Can Transformers Jump Around Right in Natural Language? Assessing Performance Transfer from SCAN


**Summary Of The Paper:**

This paperintroduce semantic linking for systematic generalization through the analysis of inductive and deductive learning from a meaningful learning perspective. They show that both prior knowledge and semantic linking play a key role in
systematic generalization, which is in line with the so-called 'meaningful learning theory'. Interesting results are attained from
SCAN to real data.

**Summary Of The Review:**

The paper has some merits but the overall claims are not well justified. The work can be improved after a major revision and narrow down the claim a bit.

---

> ### Author Response · Authors · 2021-11-18
> **Response to Reviewer yQN2**
>
> Thanks for your time and thoughtful comments. We summarize your concerns as (Q) and offer our answers (A) as follows:
>
> Q1: "domain knowledge about task structure"
>
> A1: We focus on examining models' compositional skills given semantic linking instead of introducing novel data augmentation methods. Hence, we generate variants samples given the domain knowledge (e.g., "jump\_0" is a variant of "jump") for the experiment design so that an apparent zero-shot compositional generalization can be observed. Regarding the realistic usage, we conclude data augmentation methods as inductive and deductive learning-based groups and show how models' compositional skills already promote solving real problems (Section 4). In terms of grammatical rules induction, we present how neural networks can generalize to not only variants but also new compositions (new patterns or new grammatical rules) in Difficult and Challenging settings (Section 3.4.1). To be specific, our findings demonstrate that networks can generalize to new compositions of variants, for example, "jump\_0 twice" without seeing either primitive samples with the same context (e.g., "jump twice") or primitive samples of the same length (e.g., "jump twice", "jump right", "jump left", "jump thrice", etc.). We appreciate your helpful advice, and we will leave the direction of the neural-symbolic system in future work since it is not the point for this one.
>
> Q2: "algorithmic novelty"
>
> A2: Our main contribution stands on the finding that modern neural networks can one-shot generalize to new concepts via meaningful learning as humans do, rather than a novel data augmentation to improve compositional generalization. The two common data augmentation methods help us to set up semantic linking for the purpose of our work, and we can hardly state that augmenting the training set either with samples having context (e.g., replacement augmentation) or without context (e.g., word translation) is an exciting and novel idea (Section 4). Given that, we agree with the comment that "the algorithmic novelty only combining with semantic linking is quite doubtful", but we want to stress that our findings are encouraging for followers to interpret the exceptional generalization ability of neural networks and the usage of many existing data augmentation methods from a meaningful learning perspective.
>
> Q3: "inductive and deductive processes are not very well justified"
>
> A3: Following the meaningful learning principle, we propose to build semantic linking in two ways, that is, inductive and deductive learning. They are two learning strategies for humans to learn new concepts on the basis of prior knowledge. For machines, we mimic two learning processes by involving more training samples with or without context. Thus, it is correct that "the model only uses rules to generate extra parallel data for training". However, this is not the point of this work. Instead, our experimental results on the generated dataset (Section 3.3) supports that networks are able to generalize from primitives to their variants (e.g., from "jump twice" to "jump\_0 twice"), as well as new compositions (e.g., generalize to "jump\_0 twice" without seeing "jump twice"), after training on only one augmented variant sample (Section 3.4.1).
>
> Q4: "open question and future work"
>
> A4: We share the same interests in the direction of generalization biases from artificial data sets to real ones, which is also the motivation of this work ("FROM SCAN TO REAL DATA"). There are already many works on data augmentation methods or meta-learning to improve models' compositionality in SCAN, while there are still extensive conjectures into the question of whether neural networks lack such cognitive capacity or we just do not train them appropriately or do not recognize their compositional behavior. Thanks for the suggestions, and we will revise the paper accordingly and keep exploring this direction for future work.
>
> Q5: "cognitive concept not very clear"
>
> A5: Thanks for pointing this out. To introduce meaningful learning theory to systematic generalization more friendly, we will provide examples along with definitions in Section 2.1 & 2.2, clarify the caption of Figure 1, involve the meaningful learning definition from literature in Appendix, and, as you mentioned, have an algorithm to introduce how we examine models' compositional skills in Section 3.3 in revision.

---

> > ### Comment · Reviewer_yQN2 · 2021-11-26
> > **Thanks for clarification**
> >
> > Thanks for the response. I believe it is an interesting paper but some concerns of mine still exist. Therefore, I retain my score. The paper can be improved via a major revision espacially around the formulation and theory of meaningful learning and how it fits to compositional semantic parsing tasks.

---

### Official Review · Reviewer_NJJw · 2021-11-02

**Correctness:** 4
**Technical Novelty And Significance:** 2
**Empirical Novelty And Significance:** 1
**Recommendation:** 3
**Confidence:** 5

**Details Of Ethics Concerns:**

The authors should discuss possible biases in their data augmentation.

**Main Review:**

Overall I think the presentation of the algorithm can be further improved by walking through a concrete example, especially for 2.2 and 2.3. For example, when the authors says, "getting prompt by replacing words with slot mark", it will be great if we can get a concrete example showing what's exactly the prompt. Meanwhile, the arrows in Figure 1 is kind of confusing. For example, why is the arrow from "prior knowledge" to the bottom middle box the same type of arrow as the one from the bottom middle box to "new compositions?"

My primary concerns about this paper are the following.

1. The learning setup contains very strong prior knowledge about the domain, which, sometimes can be wrong. For example, in real-world datasets because of pragmatics, "red" and "green" might not be always interchangeable: "you have a green light."
2. If we view the algorithm just as a data augmentation technique, I believe, at least for the cases studied in this paper, it has been covered by many existing techniques:
- https://arxiv.org/abs/2011.09039 This paper contains results for a number of heuristic augmentation techniques.
- https://arxiv.org/abs/1904.09545
- https://arxiv.org/abs/2010.03706 Learning to do data augmentation.
- https://aclanthology.org/2021.findings-acl.307.pdf (replacing a subtree with a new tree that has the same tag)

The authors discussed a bit about some related works in the second paragraph in Section 4, but I am still not seeing enough contribution compared with existing methods. For example, can I interpret the authors' comments on existing replacement-based augmentation as:

Existing methods use automatic ways to find possible replacements to augment the data (e.g., from a few occurrences, trying to guess that jump_0 and jump can be used interchangeably), however, if we know more about the domain, we can just write Python programs to tell the model that jump_0 and jump can be used interchangeably (this paper)?

If I'm going to be harsh, I would like to argue that in the settings studied in this paper (See Table 12 in the appendix), all these tasks can be performed simply by replacing "jump_0" with "jump" ("Houston city" with "New York city", etc) during test time. This assumes the same kind of knowledge as the authors' proposed technique.

3. I am not sure about the significance of having multiple splits in Table 2. Why is this setting meaningful? If we already know the heuristic rules, then what's the scenario where we need to exclude primitive training samples of the same length as their variant samples...?
Note that these generalization tests are all "easier" than the one we have in SCAN (e.g., jump generalization, where there is only one single example of the novel word "jump")


**Summary Of The Paper:**

This paper considers the problem of learning novel words from a few examples. The authors name their approach as "meaningful learning," which, at a high level, means that we should relate the new word with existing words. The concrete technique they proposed is to use domain-specific rules to generate new data that contains novel words based on the existing examples.

**Summary Of The Review:**

The paper is decently written and the statements are well supported. However, I failed to see the significance of the paper nor its contribution compared with other existing papers.

---

> ### Author Response · Authors · 2021-11-18
> **Response to Reviewer NJJw**
>
> Thank you for your careful reading and precious comments. We extract questions as (Q) and our answers (A) are shown as below:
>
> Q1: "Summary Of The Paper"
>
> A1: We feel sorry for the confusion regarding the purpose of our work. Here, we make it more clear as below:
> + We consider the problem of whether models can generalize compositionally given semantic linking.
> + Meaningful learning is not an approach of ours but a theory from the field of educational psychology.
> + Two data augmentation methods in our work are not novel techniques we proposed. Instead, they are tools for us to establish semantic linking. We name them inductive and deductive learning to be in line with meaningful learning.
> + Our main contribution is the finding that most models can one-shot generalize from old concepts to new ones given prior knowledge and semantic linking between primitives and their variants.
>
> Q2: "concrete example and Figure confusing"
>
> A2: Regarding Figure1, the two middle boxes show the augmented dataset used for deductive learning (upper) and inductive learning (lower). They are not two learning strategies for data augmentation but for semantic linking, that is, knowing "jump\_0" is a variant of "jump". In practice, the prior knowledge (left box) in combination with the augmented dataset (middle) is for training, and the new compositions (right box) set is for testing. Thus, we intend to draw these two learning arrows from the left box to the right box across the middle boxes. Thanks for pointing this out, and we will modify the caption of Figure 1 and explain Section 2.2 and 2.3 with specific examples in revision.
>
> Q3: "very strong domain knowledge"
>
> A3: We agree with the statement that we need domain knowldege (e.g., "jump\_0" is a variant of "jump"). However, it is not for "our proposed new data augmentation skill" (actually, we do not) but for experiments designed with toy sets. We use domain knowledge to generate variants samples so as to create an environment where we can observe the one-shot generalization more apparently followed by ablation studies (Section 3.3). In practice, the semantic relationships between primitive and their variants depend on the task. In your example, "red" is not a variant of "green" in (maybe) machine translation, but they can share the semantic connections in (maybe) part-of-speech tagging.
>
> Q4: "covered by many existing techniques"
>
> A4: We completely agree with your valuable comments if treating our works as one proposing an innovative data augmentation technique. Unfortunately, we can not accept these suggestions since data augmentation methods serve as tools for us to build semantic linking for experiment design and themselves are not novel, as noted in A1. Although there are many existing works regarding how to improve models' compositional capability through advanced data augmentation approaches, we can not put ours on the same page with them for a fair comparison due to different purposes. Again, we want to emphasize that our work investigates to which extent models can behave as humans do compositionally from a meaningful learning perspective. When it comes to your Python program argument, we agree with that we can simply replace all the "jump\_0" with its variants "jump" to achieve zero-shot learning with an accuracy of 100\% in our settings (Section 3.3). However, this is not the reason for us to stop exploring the generalization power of deep learning. As expressed in Abstract, "we hope our findings will encourage excavating existing neural networks’ potential in systematic generalization through more advanced learning schemes" rather than making the statement directly that "neural networks are inherently ineffective in such cognitive capacity".
>
> Q5: "significance of multiple splits in Table 2"
>
> A5: In Section 3.4.1 with Table 2, we remove primitive samples to weaken the prior knowledge in order to measure the role of prior knowledge in systematic generalization via meaningful learning. In Difficult scenario, we remove primitive samples sharing the same context with variants samples. As a result, models have to generalize from "jump right", for example, to "jump\_0 twice" without "jump twice" in the training set. In Challenging scenario, we filter out primitive samples of the same length as variants samples. To be specific, we remove primitive samples of length 2 from the training set, including "jump twice", "jump right", "jump left", "jump thrice", and many others. After that, models have to keep the same one-shot generalization to "jump\_0 twice". Thanks for suggestions and we will enhance our motivation for this experiment design.

---

### Official Review · Reviewer_LcKh · 2021-11-04

**Correctness:** 3
**Technical Novelty And Significance:** 2
**Empirical Novelty And Significance:** 3
**Recommendation:** 5
**Confidence:** 4

**Main Review:**

Strengths:

1. The data augmentation method is interesting, bringing semantic linking to bridge the new and old concepts. Deductive learning and inductive learning provide two learning directions for specific-to-general and general-to-specific.

2. It designs and conducts comprehensive experiments for analyzing how the semantic linking in inductive learning and deductive learning affects the models. The ablation study shows how various primitives impact the model.

Weaknesses:

1. The technical contribution is weak. It proposes the augmentation approach using the semantic link but does not study how to better use the semantic link. I would like to see more insights into how to design the model beyond these traditional sequence-to-sequence models.

2. It talks a lot about inductive learning and deductive learning but does not provide learning approaches in the deductive style.

3. Some arguments and claims are not appropriate and make it harder to understand the approaches. For example, the two methods for injecting the semantic link are actually rules with or without context. The inductive learning and deductive learning story make it ambiguous on whether it is a learning strategy or data augmentation strategy. I suggest the proper tone should be augmenting the data and then explaining the learning process with augmented data in two paradigms, but not listing the inductive/deductive learning at the beginning, which makes it very confusing.


**Summary Of The Paper:**

The paper introduces an interesting idea of improving the systematic generalization ability via meaningful learning. Through providing augmented data for inductive learning and deductive learning, the sequence-to-sequence model can be more generalizable to compositions of new concepts. It tests on real data to provide evidence of the efficacy.

**Summary Of The Review:**

The technical contribution is relatively weak and the arguments and stories of this paper make it hard to understand the specific contribution of each part.

---

> ### Author Response · Authors · 2021-11-18
> **Response to Reviewer LcKh**
>
> Thanks for your time and valuable comments. We conclude problems (Q) and reply (A) as follows:
>
> Q1: "Summary Of The Paper"
>
> A1: We want to reiterate that our work aims to evaluate models' compositional skills when semantic links between primitives and their variants exist. Thus, rather than introducing novel data augmentation tricks, our main contributions can be concluded as the statement that neural networks can one-shot generalize from primitives to variants compositionally through semantic linking. We believe our findings can, to a certain degree, explain the effectiveness of recent data augmentation methods and the generalization power of modern deep learning models in real tasks such as machine translation and semantic parsing.
>
> Q2: "the technical contribution is weak"
>
> A2: As noted in A1, our point is to study whether models can perform compositionally with the help of semantic linking. We augment the training data to build semantic linking, and the augmentation methods themselves are pretty popular and easy to use. We define them as inductive and deductive learning to follow the meaningful learning principle. From this view, we believe our findings could be helpful in further research to explore the interpretability of neural models. Sorry for the mixing, and we will improve our statements in revision. Also, we would like to thank you for your suggestions regarding how better to use the semantic link, like enhancing the model structure. We are working on this topic and will share the progress in future work.
>
> Q3: ”learning approaches“
>
> A3: The purpose of our work is to verify the extent to which neural networks can do the same as humans to generalize from old concepts to new ones systematically. In meaningful learning, inductive and deductive learning are two ways for humans to connect old concepts to new ones. As you pointed out, they can be treated as learning from variant rules with or without context in the field of data augmentation for machines (Section 2.2 & 2.3). Thus, for your questions, they are two learning strategies in meaningful learning and two data augmentation strategies in our experiments. We intend to introduce meaningful learning in this way so as to maintain the contrast between humans and machines. Again, we are sorry for blurring the primary goal of our work, and we will revise the paper accordingly.

---

### Official Review · Reviewer_PBxU · 2021-11-09

**Correctness:** 3
**Technical Novelty And Significance:** 2
**Empirical Novelty And Significance:** 2
**Recommendation:** 5
**Confidence:** 2

**Main Review:**

Strengths:
1. The interpretation of system generalization from the psychological concepts of meaningful learning and semantic linking is really interesting and reasonable.
2. The author designed extensive experiments to prove the concepts and also show the effectiveness on real data - machine translation and semantic parsing.

Weaknesses:
1. Although the author is well motivated by some psychological concepts such as meaningful learning, semantic linking or prior knowledge, I had a hard time to draw the connection between these concepts to targeted systematic generation tasks (e.g., SCAN). I think the author should give a more concise definition of these concepts, and improve the examples in Figure 1 with better explanation.
2. Although the perspective of inductive and deductive learning sounds fancy and reasonable, the introduced data augmentation methods are quite simple, and a bit frustrating.
3. It seems this paper only focuses on a specific type of generalization - lexical variants, while most of the systematic generation research focuses on compositionality. The paper seems to overclaim and not be clear at the early part of the paper.

**Summary Of The Paper:**

This paper revisits the problem of neural network's systematic generalization ability from the perspective of meaningful learning, or more specifically, semantic linking. Based on this view, they propose two data augmentation methods from either the inductive learning perspective or deductive learning perspective. They train different model variants with such augmented data. The empirical results on SCAN, GEO, and ADV show that models can behave systematically. They further group some data augmentation methods on the machine translation task and semantic parsing task into the inductive or the deductive category, and show these augmentation methods can bring benefit in real data.


**Summary Of The Review:**

This paper provides a new perspective of systematic generalization, and gives empirical results to prove that methods based on this perspective can improve model's generalization on toy and real tasks. However, the explanation of the new perspective is not clear enough, and the proposed methods look pretty simple and only solve simple generalization problems.

---

> ### Author Response · Authors · 2021-11-18
> **Response to Reviewer  PBxU**
>
> Thank you for your helpful comments. We summarize problems (Q) and provide our response (A) as follows:
>
> Q1: "Summary Of The Paper"
>
> A1: In this paper, we aim to argue that modern neural networks can one-shot generalize compositionally from old concepts to new ones conditioned on semantic linking, as concluded at the end of the Introduction. Although we define two data augmentation methods as inductive and deductive learning from the meaningful learning perspective, these two methods are not novel but serve as tools to set up semantic linking between primitives and their variants. Thus, we would like to underline that the contribution of this work does not stand on introducing data augmentation methods but exploring the interpretability of neural models in systematic generalization. Sorry for the confusion, and we will emphasize this in revision.
>
> Q2: "concise definition of psychological concepts with detailed examples"
>
> A2: We share the same concern that meaningful learning may need a detailed description so that the involvement of psychological concepts can be more friendly. Given that, we use the whole Section 2 to introduce meaningful learning in general with the support of referred literature and Appendix B on semantic linking. Thanks for pointing this out. In revision, we will clarify these concepts by involving more examples in Section 2, a more detailed explanation in Figure 1, and a more concise psychological definition in Appendix.
>
> Q3: "introduced data augmentation methods are quite simple"
>
> A3: As we stress in A1, our main contributions lie in observing successful systematic generalization with semantic linking and illustrating how this already benefits real tasks rather than introducing novel data augmentation techniques. We treat two common data augmentation methods as inductive and deductive learning to align with the meaningful learning theory, and we agree with the statement that these two methods are straightforward. With their assistance, as shown in our experiments, we can easily establish semantic linking to evaluate models' systematic generalization.
>
> Q4: "only solve a specific type of generalization - lexical variants"
>
> A4: We evaluate models' compositional generalization across a sliding scale of difficulty in Section 3.4.1. In Difficult, we remove primitive training samples sharing the same context with their variant samples. For example, models have to generalize to "jump\_0 twice" given the semantic linking between "jump" and "jump\_0" without seeing "jump twice". In Challenging, we exclude primitive training samples of the same length as their variant samples. For instance, models have to reproduce the same compositional generalization without seeing primitive training samples of length 2, including "jump twice", "jump right", "jump left", "jump thrice", and many others. Regarding variants, we take three types into consideration, namely, Lexical Variant, Co-hyponym, and Synonym (Section 2.1). We also describe each type in detail in Appendix B. In conclusion, our work investigates systematic generalization to not only new concepts but also new compositions, and we are willing to highlight this in revision.

---

### Author Response · Authors · 2021-11-22
**General Response**

We thank reviewers for their time and valuable comments and make the following revisions accordingly:

1. We revise Abstract, Introduction, and Conclusion to emphasize the topic, motivation, and contributions of our works.
2. We revise Section 2.2 and 2.3 to involve specific examples along with our definitions. We also name out the data augmentation technique used to conduct semantic linking to underline that they are not proposed by us.
3. We revise the caption of Figure 1 to make it more clear.
4. We revise Section 3.4.1 to highlight that our experiments consider not only variants but also their new compositions.
5. We reorganize the paper to fit the page limit.
6. We are willing to remove the "data augmentation" from Keywords, but it seems not allowed.

All of our revisions have been updated in OpenReview now. Thank you all!

---

### Decision · Program_Chairs · 2022-01-20

**Decision:**

Reject

**Comment:**

This paper attempts to rationalize data augmentation techniques for compositional generalization by evoking the principle of meaningful
learning which posits that learning new concepts builds on previously learned concepts (which learners already understand). So for compositional generalization, this means that a model exposed to some new concepts in the test set, should link them to known concepts which have been already attested in the training set. The links between concepts are presumed to be semantic, e.g., hyponyms, hypernyms, or lexical variants. Ideally, a model should perform semantic linking on its own, however the authors do not propose a linking mechanism. Rather they investigate data augmentation as a way of exposing a model to semantic links and then explore whether different operationalizations of semantic linking enable the model to generalize better. Inductive learning is a bottom-up approach, where links are created from general to specific concepts, whereas deductive learning is a top-down approach where links are created from specific to general concepts. Experimental results indicate that inductive learning works better.

The reviewers had the following issues with the submission (a) the technical contribution is not very strong (the idea of data augmentation is not new, although the authors' meaningful learning perspective is) (b) semantic linking seems to be able to handle only cases pertaining to lexical generalization (even though the authors include examples with structural generalization in their splits, there is no reason why semantic linking could handle these cases); (c) it would be more interesting/useful  to learn the linking than assume it is given. The authors did their best to respond to the criticism, but ultimately addressing the criticism is future work. I would also recommend to take a look at this dataset which might be useful for machine learning experiments: https://arxiv.org/abs/2105.14802